# Characterizing COVID-19 clinical phenotypes and associated comorbidities and complication profiles

**Elizabeth R. Lusczek**[1], **Nicholas E. Ingraham**[2]*, **Basil S. Karam**[3], **Jennifer Proper**[4], **Lianne Siegel**[4], **Erika S. Helgeson**[4], **Sahar Lotfi-Emran**[2], **Emily J. Zolfaghari**[5], **Emma Jones**[1], **Michael G. Usher**[6], **Jeffrey G. Chipman**[1], **R. Adams Dudley**[2,7], **Bradley Benson**[6], **Genevieve B. Melton**[1,7], **Anthony Charles**[8,9], **Monica I. Lupei**[10], **Christopher J. Tignanelli**[1,7,11]

1 Department of Surgery, University of Minnesota, Minneapolis, MN, United States of America,
2 Department of Medicine, Division of Pulmonary and Critical Care, University of Minnesota, Minneapolis, MN, United States of America, 3 Department of Surgery, Medical College of Wisconsin, Milwaukee, WI, United States of America, 4 Division of Biostatistics, School of Public Health, University of Minnesota, Minneapolis, MN, United States of America, 5 University of Minnesota Medical School, Minneapolis, MN, United States of America, 6 Department of Medicine, Division of General Internal Medicine, University of Minnesota, Minneapolis, MN, United States of America, 7 Institute for Health Informatics, University of Minnesota, Minneapolis, MN, United States of America, 8 Department of Surgery, University of North Carolina, Chapel Hill, NC, United States of America, 9 School of Public Health, University of North Carolina, Chapel Hill, NC, United States of America, 10 Department of Anesthesiology, University of Minnesota, Minneapolis, MN, United States of America, 11 Department of Surgery, North Memorial Health Hospital, Robbinsdale, MN, United States of America

☯ These authors contributed equally to this work.
* ingra107@umn.edu

## Abstract

### Purpose

Heterogeneity has been observed in outcomes of hospitalized patients with coronavirus disease 2019 (COVID-19). Identification of clinical phenotypes may facilitate tailored therapy and improve outcomes. The purpose of this study is to identify specific clinical phenotypes across COVID-19 patients and compare admission characteristics and outcomes.

### Methods

This is a retrospective analysis of COVID-19 patients from March 7, 2020 to August 25, 2020 at 14 U.S. hospitals. Ensemble clustering was performed on 33 variables collected within 72 hours of admission. Principal component analysis was performed to visualize variable contributions to clustering. Multinomial regression models were fit to compare patient comorbidities across phenotypes. Multivariable models were fit to estimate associations between phenotype and in-hospital complications and clinical outcomes.

### Results

The database included 1,022 hospitalized patients with COVID-19. Three clinical phenotypes were identified (I, II, III), with 236 [23.1%] patients in phenotype I, 613 [60%] patients

**Data Availability Statement:** All data cannot be shared publicly because these data contain private electronic healthcare record data from M Health Fairview and is subject to HIPAA laws which

restrict sharing of data without data use agreements in place. In accordance with HIPPA regulations, a de-identified dataset has been included with this publication.

**Funding:** 1. NIH National Heart, Lung, and Blood Institute T32HL07741 (NEI) 2. This research was supported by the Agency for Healthcare Research and Quality (AHRQ) and Patient-Centered Outcomes Research Institute (PCORI), grant K12HS026379 (CJT) and the National Institutes of Health's National Center for Advancing Translational Sciences, grant UL1TR002494. 3. NIH National Heart, Lung, and Blood Institute T32HL129956 (JP, LS) The funders had no role in study design, data collection and analysis, decision to publish, or preparation of the manuscript.

**Competing interests:** The authors have declared that no competing interests exist.

in phenotype II, and 173 [16.9%] patients in phenotype III. Patients with respiratory comorbidities were most commonly phenotype III (p = 0.002), while patients with hematologic, renal, and cardiac (all p<0.001) comorbidities were most commonly phenotype I. Adjusted odds of respiratory, renal, hepatic, metabolic (all p<0.001), and hematological (p = 0.02) complications were highest for phenotype I. Phenotypes I and II were associated with 7.30-fold (HR:7.30, 95% CI:(3.11–17.17), p<0.001) and 2.57-fold (HR:2.57, 95% CI:(1.10–6.00), p = 0.03) increases in hazard of death relative to phenotype III.

## Conclusion

We identified three clinical COVID-19 phenotypes, reflecting patient populations with different comorbidities, complications, and clinical outcomes. Future research is needed to determine the utility of these phenotypes in clinical practice and trial design.

## Introduction

The coronavirus disease 2019 (COVID-19), a disease caused by the severe acute respiratory syndrome coronavirus-2 (SARS-CoV-2), has infected over 18 million and led to over 700,000 deaths since first appearing in late 2019 [1]. Researchers are rapidly attempting to understand the natural history of and immune response to COVID-19 [2]. Despite intense research since the arrival of this novel coronavirus [3], only one pharmaco-therapeutic agent, dexamethasone, has been associated with reduced mortality in at-risk individuals [4]. COVID-19 results in a constellation of symptoms, laboratory derangement, immune dysregulation, and clinical complications [5].

Emergency department presentation varies widely, suggesting distinct clinical phenotypes exist and, importantly, it is likely these distinct phenotypes respond differently to treatment. To illustrate, two early phenotypes of respiratory failure likely exist in COVID-19. A classic ARDS phenotype exists with poorly compliant lungs and poor gas exchange; however, a phenotype with normal lung compliance also exists in COVID-19 and is hypothesized to be driven by shunting secondary to pulmonary microthrombi [6, 7]. An intricate, multidimensional view is required to adequately understand the disease and account for the variation in clinical outcomes. Furthermore, patients could benefit from phenotype-specific medical care, which may differ from established standards of care.

Despite this need, few studies have characterized COVID-19 clinical phenotypes and evaluated their association with complications and clinical outcomes. The aim of this study was to characterize clinical phenotypes in COVID-19 according to disease-system factors using electronic health record (EHR) data pooled from 14 U.S. Midwest hospitals between March 7, 2020 and August 25, 2020.

## Materials and methods

### Data collection

The data source for this study included EHR reports from 14 U.S. Midwest hospitals and 60 primary care clinics across Minnesota. The healthcare system includes an academic quaternary center along with community hospitals all capable of providing critical care. Patient and hospital-level data were available for 7,538 patients with PCR-confirmed COVID-19. Of these, 1,022 required hospital admission and were included in this analysis. The database included all

comorbidities reported since March 29, 1997 for each patient and prior to their COVID-19 diagnosis. The database also included home medications, laboratory values, clinic visits, social history, and patient demographics (age, gender, race/ethnicity, language spoken, zip code, socioeconomic status indicators). Race/ethnicity are self-reported. For each COVID-19 hospitalization the database included all laboratory values, vitals, orders, medications, complications, length of stay, and hospital disposition. State death certificate data was linked with the database to enable capture of out-of-hospital death. Additionally, the database allowed linkage across the 14 hospitals, facilitating the tracking of transfers.

The study was approved by all hospitals within the MHealth Fairview system which includes ethical approval by the University of Minnesota institutional review board. All patients have the option to opt-out of research upon establishing care within the MHealth Fairview healthcare system. Data is aggregated through the University of Minnesota's centralized informatics center and de-identified prior to analysis. Data were pooled across different electronic health records (EHRs) utilizing a unique patient identifier to account for health care encounters across systems. This study was approved by the University of Minnesota institutional review board (STUDY00001489), which provided a waiver of consent for this study.

## Participants

Patient-level data were obtained from the COVID-19 database from March 7, 2020 to August 25, 2020. The inclusion criterion was as follows: PCR-positive COVID-19 test requiring inpatient hospital admission to one of the 14 hospitals providing data. No hospitalized patients were excluded in this analysis to maximize generalizability. Follow-up data were available for a minimum of two weeks following admission for all patients.

## Clinical variables for phenotyping

We selected 33 variables for clustering based on their association with COVID-19 mortality, known COVID-19 pathophysiology, and presence in the database (no more than 50% missingness) [8–11]. The following variables were included: age, body mass index (BMI), heart rate, respiratory rate, oxygen saturation, pulse pressure, systolic blood pressure, total protein, red cell distribution width, mean corpuscular volume, alkaline phosphatase, calcium, anion gap, bicarbonate, hematocrit, aspartate aminotransferase, glucose, absolute monocyte count, absolute neutrophil count, absolute lymphocyte count, white blood cell count, platelet, albumin, bilirubin, international normalized ratio (INR), lactate dehydrogenase, potassium, sodium, D-dimer, hemoglobin, C-reactive protein (CRP), creatinine, and gamma gap. For each variable we selected the first recorded value within the first 72 hours of the emergency department (ED) presentation that ultimately resulted in their hospitalization.

## Comorbidities

We selected 68 comorbidities documented for each patient from March 29, 1997 preceding their COVID-19 hospital admission in their electronic health record (S1 Table). All comorbidities were identified based on ICD-9, ICD-10, or problem list documentation within the electronic health record. An indicator variable was created for each comorbidity to denote the presence of the selected ICD-9, ICD-10, or problem list documentation at any time in the medical record. To facilitate analysis, comorbidities were grouped by organ system into the following categories: cardiac, respiratory, hematologic, metabolic, renal, hepatic, autoimmune, cancer, and cerebrovascular disease.

## Complications and clinical outcomes

We selected 30 in-hospital complications measured during each patient's hospital stay for COVID-19 categorized into the following systems: cardiovascular, respiratory, hematologic, renal, hepatic, metabolic, and infectious (S2 Table). If applicable, complications could span multiple organ system variables. For example, ventilator associated pneumonia was included in both infectious and respiratory complications. Additional clinical outcomes included hospital length of stay (LOS), need for intensive care unit (ICU) admission, need for mechanical ventilation, and mortality. Mortality was defined as any in-hospital or out-of-hospital death based on death certificate data. All complications and outcomes were followed for a minimum of 2 weeks following hospital admission.

## Statistical analysis

The overall rate of missingness of the 33 variables used for phenotyping, which included the first vitals and labs recorded for each inpatient within 72 hours of admission, was 19% (range 0% - 50%). We imputed missing values using multivariate imputations by chained equations implemented with the *mice* package (v.3.10.0) [12, 13]. Data were log-transformed before imputing missing values with predictive mean matching. A total of 40 imputed datasets were generated. The *diceR* package (v.1.0.0) [14] was used to perform k-means-based consensus clustering on each imputed dataset using 80% subsamples and 1,000 iterations. We considered grouping patients into 2–7 phenotypes and determined the optimal number was 3 by evaluating the consensus cumulative distribution function (CDF) plot, the delta area plot, and the consensus matrix heatmap. These figures were generated using the consensus clustering results for each imputed dataset, and all figures were qualitatively similar across datasets. For visualization purposes, these images are provided for a randomly selected imputed dataset in S1–S4 Figs. The final assignment of each patient into one of the three phenotypes was determined by majority voting across the 40 consensus clustering results. Principal component analysis (PCA) was performed on the average covariance matrix to visualize the relationships among the three phenotypes and assess variable contributions [15].

Continuous variables were summarized using the median and interquartile range (IQR) and compared across phenotypes using a Kruskal-Wallis test. Categorical characteristics and outcomes were summarized using counts and proportions and compared across phenotypes using a Pearson's chi-squared test or Fisher's exact test. Multinomial regression models were fit to further compare patient comorbidities across phenotype classification.

We next evaluated the relationship between phenotype and subsequent outcomes using both unadjusted and adjusted models. The adjusted models included sex [16, 17], race and ethnicity (white, Black, Asian, Hispanic, other, not reported) [18], and Elixhauser Comorbidity Index [19], since these are known risk factors for the outcomes of interest and were not included in the clustering analysis. The associations between phenotype and complications, ICU admission and need for mechanical ventilation, were estimated using logistic regression models. Mortality was compared across phenotypes using Cox proportional hazard models and patients were censored at the last date of data collection, August 25, 2020. Hospital length of stay was compared across phenotypes using negative binomial regression models. The primary negative binomial model included individuals who died during hospitalization for whom length of stay was defined as the number of days until death. We performed a sensitivity analysis to assess the impact of mortality as a competing risk by refitting the length of stay model after removing the 127 patients who died. Two-sided p-values < 0.05 were considered statistically significant. P-values were not adjusted for multiple comparisons. Visualizations of comorbidities, complications, and outcomes by clinical phenotype were performed using the

*circlize* package for R [20]. Comorbidities and complications were grouped into separate organ systems and the prevalence of each complication/comorbidity type was calculated as a percentage for each phenotype. All analyses were conducted using R version 3.6.3 [21] and Stata version 16.1 (StataCorp).

## Results

The database included 1,022 patients requiring hospital admission with COVID-19. Among these patients, the median age was 62.1 [IQR: 45.9, 75.8] years; 481 [48.6%] male, 412 [40.3%] required ICU admission. Additionally, 437 [46.7%] were white, 188 [20.1%] were Black, 159 [17.0%] were Asian, 103 [11.0%] were Hispanic, 20 [2.1%] reported other race, and 28 [2.9%] did not report. Three clinical phenotypes were identified (I, II, III); 236 [23.1%] patients had phenotype I, 613 [60%] patients had phenotype II, and 173 [16.9%] patients had phenotype III.

### Variable contributions to clustering

The first two principal components (PCs) from PCA were used to visualize the relationship between phenotypes. PC1 and PC2 captured approximately 11% and 9% of the variance in the clustering variables, respectively. Thirteen components were needed to explain 70% of the variance (S5 Fig). While phenotypes II and III overlay substantially, phenotype I is more clearly defined in the right-hand side of the score plot of the first two principal components (Fig 1). Notably, this figure shows that distinctions between phenotypes are primarily driven by variation in PC1 as opposed to PC2. The variable contributions to PC1 (S6 Fig) demonstrate that the largest contributors to the variation in PC1 are from lactate dehydrogenase (LDH), absolute neutrophil count, and D-dimer. These variables therefore prominently contribute to separating the three phenotypes as shown in the biplot (Fig 2). Univariate tests showed that LDH, D-dimer, and neutrophil count are highest in phenotype I. Other variables influential to phenotype clustering are white cell count (highest in I), C-reactive protein (highest in I), albumin (highest in III), aspartate aminotransferase (highest in I), bilirubin (highest in I), and oxygen saturation (highest in III).

### Phenotype characteristics

Differences across phenotypes with respect to patient demographics, admission vitals and labs, complications, comorbidities, and clinical outcomes are presented in Table 1. Patients with phenotype I were older than patients in phenotypes II and III (67.2 [52.9, 79.0] years vs. 60.9 [45.9, 75.4] and 58.6 [34.8, 71.3] years respectively, p < 0.001). Patients with phenotype III were more often female than patients with phenotype I or II (57.6% vs. 41.6% and 53.4%, respectively, p = 0.002). Patients with phenotype I were less likely to be white (38.8% vs. 45.6% vs. 60.7%, respectively, p = 0.002) and more likely to be non-English speaking (47.9% vs. 39.2% vs. 23.7%, respectively, p <0.001). There were no statistically significant differences in BMI or socioeconomic status, as measured using the area deprivation index, between phenotypes (Table 1). Patients that presented with phenotype III had a more frequent history of smoking, alcohol abuse, and neutropenia. Patients that presented with phenotype II had a less frequent history of hepatic disease than phenotypes I or III (Table 1).

When grouping comorbidities by organ system, cardiac (p <0.001), respiratory (p = 0.002), hematologic (p <0.001), and renal (p <0.001) comorbidities were found to be significantly associated with phenotype. Cancer, hepatic, autoimmune, cerebrovascular, and metabolic comorbidities were not significantly associated with phenotype (Table 1, S7 Fig). Based on the estimated relative risk ratios, patients with renal (RRR 2.35; 95% CI 1.5–3.67; p <0.001), hematologic (RRR 2.64; 95% CI 1.75–3.98; p <0.001), and cardiac comorbidities (RRR 2.65; 95% CI:

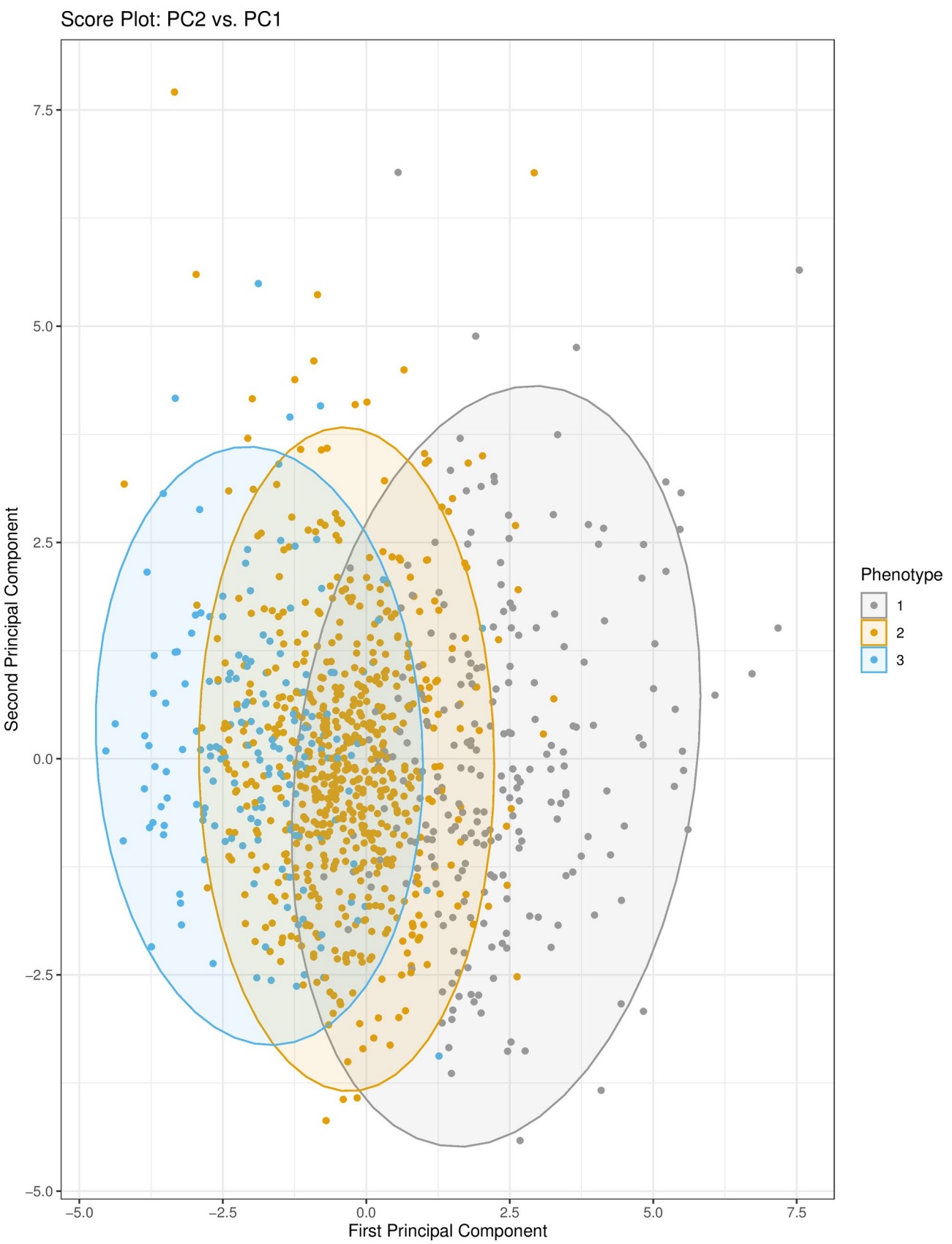

**Fig 1. Score plot: PC2 vs. PC1.** The principal component scores for PC1 and PC2 are plotted. Each point represents a patient in the dataset. Colors represent the cluster (phenotype) that the patient was assigned to by consensus clustering. Ellipses around each cluster/phenotype specify 95% confidence intervals, assuming a bivariate normal distribution. Abbreviations: PC1 (principal component 1); PC2 (principal component 2).

1.68–4.17; p <0.001) were more likely to have phenotype I vs. III (Fig 3). Patients with respiratory comorbidities were 0.47 (95% CI: 0.31–0.72; p <0.001) times as likely to have phenotype I vs. III and 0.74 (95% CI: 0.52–1.04 p = 0.09) times as likely to have phenotype II vs. III (Fig 3).

## Association between phenotype and clinical outcomes

Clinical phenotypes I and II were associated with increased odds of respiratory (I: OR: 2.98, 95% CI 1.58–5.59; II: OR: 2.32, 95% CI: 1.29–4.17; p<0.001), renal (I: OR: 7.04, 95% CI 3.11–15.9; II: OR: 2.57, 95% CI: 1.15–5.74; p <0.001), and metabolic (I: OR: 4.85, 95% CI: 2.78–8.45; II: OR: 2.57, 95% CI: 1.52–4.34; p <0.001) complications, compared to phenotype III after adjusting for sex, race, and Elixhauser Comorbidity Index (S3 Table). There was a trend towards increased odds of hematologic complications among patients with phenotype I (I: OR: 2.11, 95% CI: 0.99–4.48, p = 0.05) compared to III. Phenotype was associated with hepatic complications (p <0.001); however, while phenotype I was associated with a 8.35-fold (OR: 8.35, 95% CI: 1.93–36.11, p < 0.001) increase in the odds of hepatic complication, phenotype II did not differ significantly from phenotype III (OR: 0.56, 95% CI: 0.10–3.09, p = 0.51). This is not surprising since only 4 individuals in phenotype II and 2 in phenotype III experienced hepatic complications during hospitalization (Table 1). Phenotype was also significantly associated with the rate of infectious complications (p <0.001) for phenotype 1 (OR 2.57, 95% CI 1.57–4.21; <0.001) but not did not reach statistical significance for phenotype 2 (OR 1.51, 95% CI 0.96–2.38; p = 0.07) (S3 Table and S8 Fig).

Clinical phenotypes differed in odds of ICU admission (p <0.001) and mechanical ventilation (p <0.001), hospital LOS (p <0.001), and risk of mortality (<0.001) on adjusted analysis which accounted for sex, race, and Elixhauser Comorbidity Index (Table 2, S9 Fig). Controlling for these risk factors and compared to phenotype III, phenotypes I and II were associated with 7.88-fold (OR: 7.88, 95% CI: 4.65–13.37) and 2.32-fold (OR: 2.32, 95% CI: 1.46–3.68) increases in the odds of ICU admission, respectively. Phenotypes I and II were associated with 25.59-fold (OR: 25.59, 95% CI: 7.69,-85.17) and 7.45-fold (OR: 7.45, 95% CI: 2.27–24.43) increases in the odds of requiring mechanical ventilation. Phenotypes I and II were associated with 1.74-fold (IRR: 1.74, 95% CI: 1.45–2.10, p<0.001) and 1.22-fold (IRR: 1.22, 95% CI: 1.05–1.43, p = 0.01) increases in hospital LOS. Phenotype I was associated with a 7.30-fold (HR: 7.30, 95% CI: 3.11–17.17, p <0.001) increase in risk of mortality, and Phenotype II had a 2.57-fold (HR: 2.57, 95% CI: 1.10–6.00, p = 0.03) increase in the hazard of death compared to Phenotype 3. We performed a sensitivity analysis to assess the impact of mortality as a competing risk by fitting the LOS model before and after removing the 127 patients who died. The estimated effect sizes were similar between these two models (S4 Table). Table 2 includes the LOS model with only survivors. S4 Table shows the home medications and Day 5 labs of the three identified phenotypes (S5 Table).

## Discussion

This is one of the first studies to report on clinical phenotypes associated with COVID-19. We identified three clinical phenotypes for patients with COVID-19 on hospital presentation. Most patients presented with phenotype II, which is associated with a moderate course and an approximately 10% mortality. A subset of patients presented with the more severe phenotype I, which is associated with a staggering 27% mortality. Patients with cardiac, hematologic, and

PCA Biplot: PC2 vs. PC1

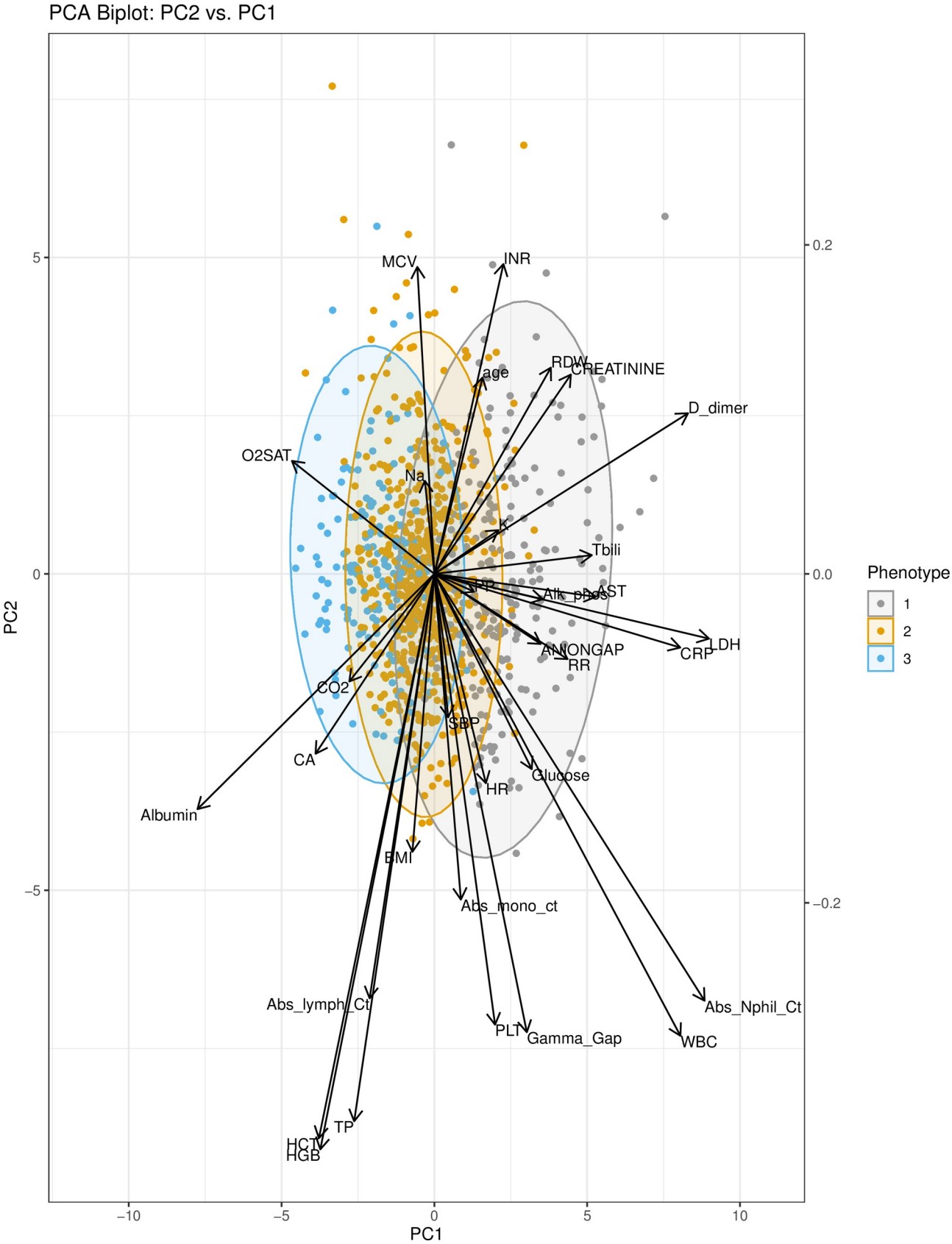

**Fig 2. PCA biplot: PC2 vs. PC1.** The scores (points) and loadings (arrows) of PC1 and PC2 are plotted for each patient and variable in the model. 95% confidence ellipses for the scores are shown. The biplot facilitates interpretation of the scores and loadings, assigning context to the variables which prominently contribute to the phenotypes. Abbreviations: PC1 (principal component 1); PC2 (principal component 2); PCA (principal component analysis); Abs_Nphil_Ct (absolute neutrophil count); LDH (lactate dehydrogenase); CRP (C-reactive protein); WBC (white blood cell count); HCT (hematocrit); HGB (hemoglobin); Tbili (total bilirubin); RDW (red cell distribution width); AST (aspartate aminotransferase); Alk_phos (alkaline phosphatase); RR (respiratory rate); CA (calcium); TP (total protein); INR (internal normalized ratio of prothrombin time); CO2 (carbon dioxide); K (potassium); O2SAT (oxygen saturation); BMI (body mass index); PLT (platelet); PP (pulse pressure); Na (sodium); SBP (systolic blood pressure); Abs_mono_ct (absolute monocyte count); MCV (mean corpuscular volume).

renal comorbidities were most likely to be characterized by phenotype I. Surprisingly, respiratory comorbidities appeared less related to phenotypes I or II and were most associated with phenotype III, which had the most indolent course. Despite this indolent course, patients with phenotype III had the highest rate of readmission which is likely in part due to the high survival rate. This also suggests patients with pre-existing respiratory comorbidities, while not at highest risk for mortality, may be at highest risk for long term sequalae following COVID-19. Patients that presented with phenotype I were most associated with the development of respiratory, hematologic, renal, metabolic, hepatic, and infectious complications. Surprisingly, cardiovascular complications did not significantly differ between phenotypes.

Elucidating patient risk factors and severe COVID-19 disease markers may allow early treatment implementation that may improve the patient's outcome. Multiple studies have documented COVID-19 risk factors; however, most have done so from a homogenous lens. For example, a prospective cohort study from New York City identified that the most considerable risks for hospital admission were age, male sex, heart failure, chronic kidney disease, and high BMI [22]. A large observational study conducted in the UK reported that increasing age, male gender, comorbidities such as cardiac disease, chronic lung disease, chronic kidney disease, and obesity were associated with higher mortality in COVID-19 positive patients admitted to the hospital.[14] A study from China found that increased odds of in-hospital death due to COVID-19 were associated with older age, higher sequential organ failure assessment (SOFA) score and D-dimers > 1.0 μg/mL on admission [23]. Another retrospective study reported that patients with severe COVID-19 disease and diabetes had increased leucocytes, neutrophils count, and increased C-reactive protein (CRP), D-dimers, fibrinogen levels [24]. A systematic review and meta-analysis found that the biomarkers associated with increased mortality include higher CRP, higher D-dimers, increased creatinine, and lower albumin levels [25]. However it is well known that patients do not have a singular natural history of disease. Multiple studies including this study found that only half of patients suffer a primarily respiratory disease [26, 27]. Patients suffer a constellation of cardiovascular, hematologic, renal, or hepatic progression of disease following COVID-19. It is likely patient baseline risk factors related to the virus [28], home medications [16, 29], genetic predisposition [30], race/ethnicity [18], and other factors predispose patients to one of the various clinical manifestations and natural history of COVID-19.

Treatment of hospitalized patients should be tailored based on the clinical courses most likely for a patient given their *a priori* risk. For example, phenotypes with a higher risk of thrombotic events, may benefit from more aggressive anticoagulation. Phenotypes more prone to infectious complications, may benefit from more targeted immunomodulation instead of broad and systemic steroid therapy. A key first step to evaluate these treatment decisions is to characterize and describe clinical phenotypes requiring hospitalization. In this analysis we identified three clinical phenotypes for patients that required hospitalization for COVID-19. Few studies to date have attempted to elucidate clinical phenotypes. One study attempted to characterize clinical phenotypes at ICU admission using a dataset of 85 critically ill patients [31]. Similar to our analysis, they identified three distinct clinical phenotypes. Their

**Table 1. Baseline demographics, comorbidities, and clinical outcomes of hospitalized COVID-19 patients with clinical phenotypes I, II, and III.**

| | Phenotype I | Phenotype II | Phenotype III | P-value |
|---|---|---|---|---|
| | N = 236 | N = 613 | N = 173 | |
| **Demographics** | | | | |
| **Age (years)** | 67.2 (52.9–79.0) | 60.9 (45.9–75.4) | 58.6 (34.8–71.3) | <0.001 |
| **Male** | 132 (58.4%) | 277 (46.6%) | 72 (42.4%) | 0.002 |
| **Race / Ethnicity** | | | | 0.002 |
| White | 81 (38.8%) | 257 (45.6%) | 99 (60.7%) | |
| Black | 53 (25.4%) | 105 (18.7%) | 30 (18.4%) | |
| Asian | 39 (18.7%) | 101 (17.9%) | 19 (11.7%) | |
| Hispanic | 26 (12.4%) | 66 (11.7%) | 11 (6.7%) | |
| Declined | 3 (1.4%) | 22 (3.9%) | 3 (1.8%) | |
| Other | 7 (3.3%) | 12 (2.1%) | 1 (0.6%) | |
| **Non-English Speaking** | 113 (47.9%) | 240 (39.2%) | 41 (23.7%) | <0.001 |
| **National ADI** | 44.5 (25.0–56.0) | 43.0 (25.0–56.0) | 37.0 (26.0–62.0) | 0.76 |
| **BMI (kg/m$^2$), mean (SD)** | 29.5 (8.9) | 30.8 (8.2) | 30.4 (13.4) | 0.21 |
| **Smoker** | 9 (3.8) | 44 (7.2) | 18 (10.4) | 0.03 |
| **Alcohol abuse** | 14 (5.9) | 47 (7.7) | 28 (16.2) | <0.001 |
| **Comorbidities** | | | | |
| **Elixhauser Comorbidity Index** | 7.0 (4.0–10.0) | 5.0 (3.0–9.0) | 5.0 (2.0–8.0) | <0.001 |
| **Cardiac** | 194 (82.2%) | 428 (69.8%) | 110 (63.6%) | <0.001 |
| **Respiratory** | 55 (23.3%) | 198 (32.3%) | 68 (39.3%) | 0.002 |
| **Hematologic** | 127 (53.8%) | 220 (35.9%) | 53 (30.6%) | <0.001 |
| **Metabolic** | 175 (74.2%) | 477 (77.8%) | 121 (69.9%) | 0.08 |
| **Renal** | 92 (39.0%) | 170 (27.7%) | 37 (21.4%) | <0.001 |
| **Hepatic** | 46 (19.5%) | 82 (13.4%) | 25 (14.5%) | 0.08 |
| **Autoimmune** | 40 (16.9%) | 126 (20.6%) | 23 (13.3%) | 0.07 |
| **Cancer** | 29 (12.3%) | 73 (11.9%) | 16 (9.2%) | 0.58 |
| **Cerebrovascular disease** | 52 (22.0%) | 106 (17.3%) | 33 (19.1%) | 0.28 |
| **Blood Type O** | 72 (42.4%) | 158 (39.0%) | 39 (37.5%) | 0.67 |
| **In-hospital Complications** | | | | |
| **Cardiovascular** | 16 (6.8%) | 46 (7.5%) | 13 (7.5%) | 0.93 |
| **Respiratory** | 49 (20.8%) | 104 (17.0%) | 14 (8.1%) | 0.002 |
| **Hematologic** | 27 (11.4%) | 35 (5.7%) | 10 (5.8%) | 0.01 |
| **Renal** | 54 (22.9%) | 60 (9.8%) | 7 (4.0%) | <0.001 |
| **Metabolic** | 85 (36.0%) | 141 (23.0%) | 18 (10.4%) | <0.001 |
| **Hepatic** | 21 (8.9%) | 4 (0.7%) | 2 (1.2%) | <0.001 |
| **Infectious** | 76 (32.2%) | 134 (21.9%) | 27 (15.6%) | <0.001 |
| **Clinical Outcomes** | | | | |
| **ICU Admission** | 158 (66.9%) | 220 (35.9%) | 34 (19.7%) | <0.001 |
| **Mechanical Ventilation** | 98 (41.5%) | 88 (14.4%) | 4 (2.3%) | <0.001 |
| **Hospital Readmission** | 6 (2.5%) | 29 (4.7%) | 14 (8.1%) | 0.03 |
| **ECMO** | 7 (3.0%) | 1 (0.2%) | 0 (0.0%) | <0.001 |
| **In- or Out of hospital mortality** | 63 (26.7%) | 57 (9.3%) | 7 (4.0%) | <0.001 |
| **Admission Vitals and Labs** | Phenotype I | Phenotype II | Phenotype III | P value |
| **Heart rate** (mean (SD)) | 96.17 (20.82) | 93.93 (19.35) | 90.16 (22.3) | 0.01 |
| **Respiratory rate** | 22.0 (18.0–28.0) | 20.0 (18.0–23.0) | 18.0 (16.0–20.0) | <0.001 |
| **Oxygen saturation** | 94.0 (89.0–97.0) | 95.0 (92.0–97.0) | 97.0 (95.0–99.0) | <0.001 |
| **Pulse pressure** | 55.0 (43.5–70.5) | 53.0 (43.0–68.0) | 51.0 (40.0–62.0) | 0.02 |

*(Continued)*

**Table 1.** (Continued)

|  | Phenotype I | Phenotype II | Phenotype III | P-value |
|---|---|---|---|---|
|  | N = 236 | N = 613 | N = 173 |  |
| **SBP** (mean (SD)) | 133.29 (27.14) | 132.46 (23.54) | 134.10 (26.26) | 0.72 |
| **Total protein** | 6.5 (5.9–7.0) | 6.7 (6.20–7.2) | 6.6 (6.2–7.1) | 0.01 |
| **Red cell distribution width** | 14.1 (13.2–15.4) | 13.5 (12.9–14.7) | 13.5 (12.8–14.6) | <0.001 |
| **Mean corpuscular volume** | 90.0 (86.0–94.0) | 89.0 (85.0–93.0) | 92.0 (88.0–95.3) | <0.001 |
| **Alkaline phosphatase** | 88.0 (67.5–129.0) | 71.0 (55.5–92.0) | 72.0 (58.-88.0) | <0.001 |
| **Calcium** | 8.10 (7.6–8.5) | 8.30 (8.0–8.7) | 8.40 (8.1–8.9) | <0.001 |
| **Anion gap** | 9.0 (7.0–12.0) | 8.0 (6.0–10.0) | 7.0 (6.0–9.0) | <0.001 |
| **CO2** | 23.25 (21.0–26.0) | 24.0 (22.0–27.0) | 25.0 (23.0–27.8) | <0.001 |
| **Hematocrit** | 36.40 (32.3–40.2) | 37.60 (33.6–41.1) | 38.45 (35.7–41.5) | <0.001 |
| **Aspartate aminotransferase** | 55.0 (38.0–95.0) | 35.0 (24.0–53.0) | 29.0 (20.0–44.0) | <0.001 |
| **Glucose** | 122.0 (101.0–165.0) | 112.0 (96.0–149.5) | 104.0 (91.0–126.5) | <0.001 |
| **Absolute monocyte count** | 0.40 (0.3–0.8) | 0.40 (0.3–0.6) | 0.50 (0.3–0.7) | <0.001 |
| **Platelets** | 206.0 (160.0–290.0) | 190.0 (149.0–243.0) | 196.0 (142.5–247.5) | 0.01 |
| **Albumin** | 2.40 (2.0–2.7) | 2.80 (2.5–3.1) | 3.10 (2.8–3.4) | <0.001 |
| **Bilirubin** | 0.70 (0.4–1.1) | 0.40 (0.3–0.6) | 0.40 (0.3–0.6) | <0.001 |
| **INR** | 1.11 (1.03–1.28) | 1.06 (0.99–1.17) | 1.08 (0.98–1.21) | 0.001 |
| **Lactate dehydrogenase** | 460.5 (380.0–562.8) | 308.0 (249.0–394.0) | 231.0 (180.0–293.5) | <0.001 |
| **Potassium** | 4.0 (3.6–4.3) | 3.80 (3.6–4.2) | 3.80 (3.6–4.2) | 0.101 |
| **Sodium** | 137.5 (134.0–141.0) | 137.0 (135.0–139.0) | 138.0 (136.0–140.0) | 0.003 |
| **D-dimer** | 3.08 (1.71–5.57) | 0.87 (0.59–1.27) | 0.60 (0.36–1.05) | <0.001 |
| **Hemoglobin** | 11.90 (10.5–13.1) | 12.20 (10.7–13.5) | 12.40 (11.3–13.7) | 0.01 |
| **C-reactive protein** | 157.0 (102.0–244.0) | 89.0 (55.0–134.8) | 12.0 (5.0–20.0) | <0.001 |
| **Creatinine** | 1.06 (0.77–1.62) | 0.84 (0.69–1.13) | 0.80 (0.68–1.03) | <0.001 |
| **Absolute neutrophil count** | 8.05 (5.75–11.42) | 4.20 (3.0–6.0) | 2.90 (1.8–4.3) | <0.001 |
| **Absolute lymphocyte count** | 0.90 (0.6–1.3) | 0.90 (0.7–1.3) | 1.30 (0.9–1.7) | <0.001 |
| **WBC** | 8.74 (5.68–15.42) | 4.50 (3.0–6.71) | 2.36 (1.31–3.77) | <0.001 |
| **Gamma Gap** | 9.80 (7.2–13.2) | 5.90 (4.3–7.6) | 4.90 (3.9–7.3) | <0.001 |

Table 1 presents summary statistics of patient demographics, comorbidities, in-hospital complications, clinical outcomes, and admission vitals and labs for each clinical phenotype (I, II, III). Admissions vitals and labs were used to create the phenotypes. Categorical variables are presented as count (%). Continuous variables are presented as median (interquartile range) unless otherwise specified.

Abbreviations: ADI, area deprivation index; BMI, body mass index; INR, internal normalized ratio of prothrombin time; ECMO, Extracorporeal membrane oxygenation; ICU, Intensive Care Unit

low mortality cluster which they called cluster 1 was very similar to our phenotype III with a predominance of females, lower mortality rate, lower D-dimer and CRP levels. Similarly, their high mortality cluster was predominantly male, with elevated inflammation markers on ICU presentation. In this study, we not only characterized three clinical phenotypes, but extended findings outside of the ICU by characterizing the association of comorbidities with clinical phenotype and the association of clinical phenotypes with in-hospital complication and clinical outcomes.

Phenotype I can be termed the "Adverse phenotype" and was associated with the worst clinical outcomes. Lactate dehydrogenase (LDH), absolute neutrophil count, D-dimer, aspartate aminotransferase (AST), and C-reactive protein (CRP) were most influential in phenotype I determination. The strong association of red cell distribution width (RDW) with phenotype I was interesting. RDW was strongly associated with genetic age which is hypothesized to be a

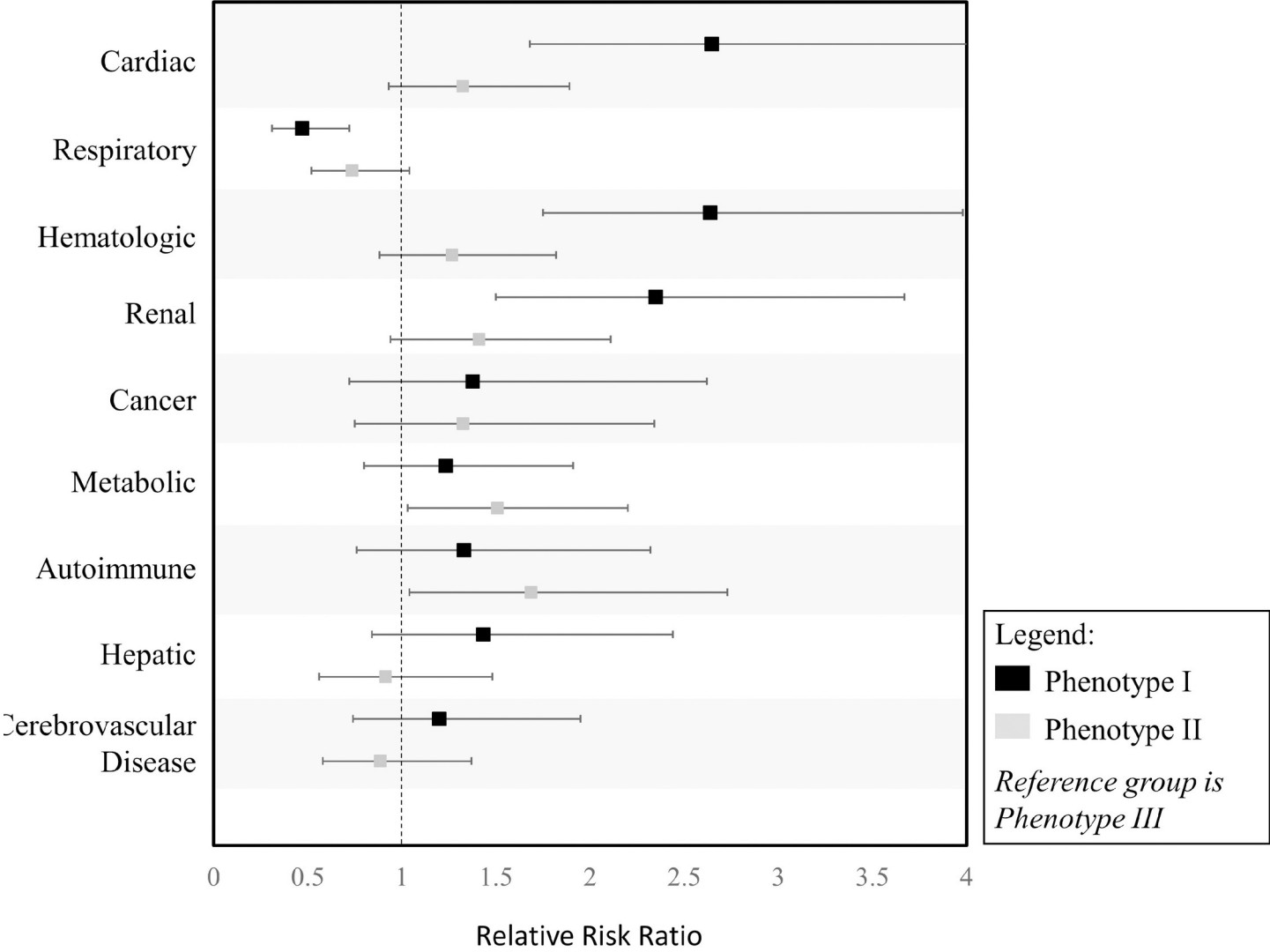

**Fig 3. Relative risk ratio of comorbidities to clinical phenotypes.** Relative Risk ratios of comorbidities of phenotypes I and II compared to the reference group phenotype III.

risk factor in COVID-19 [30]. As people age, variability in red blood cell volumes increases. Similarly, Gamma Gap, a marker of immunoglobulin levels, was elevated in all three phenotypes (median > 3.5) [32]. However, patients with clinical phenotype I were noted to have the largest increase in Gamma Gap. In this scenario elevated Gamma Gap was likely an indicator of systemic inflammation and has been associated in other inflammatory disease processes with prognosis. Other groups have previously reported on the importance of the Absolute Neutrophil to Absolute Lymphocyte count, here we noted that ANC/ALC was lowest for phenotype III and highest for phenotype I, in line with previous reports. Patients with cardiac, hematologic, and renal comorbidities were most prone to develop phenotype I. Phenotype I was associated with numerous complications (hematologic, hepatic, metabolic, renal, respiratory, and infectious) when compared to other phenotypes. It is interesting to note despite a higher rate of baseline cardiac comorbidities phenotype I was not associated with increased cardiac complications. Beyond the pathophysiologic differences, it is important to note the higher proportion of non-White and non-English speaking patients in phenotype I. Moreover,

**Table 2. Association of clinical phenotype with clinical outcome.**

| In- and Out- of Hospital Mortality *(Cox PH)* | HR | 95% CI | P value |
|---|---|---|---|
| Mortality | | | <0.001 (LR test) |
| Phenotype I | 7.30 | 3.11–17.17 | <0.001 |
| Phenotype II | 2.57 | 1.10–6.00 | 0.03 |
| **Binary Outcomes** *(Logistic Regression)* | OR | 95% CI | P value |
| ICU Admission | | | <0.001 (LR test) |
| Phenotype I | 7.88 | 4.65–13.37 | <0.001 |
| Phenotype II | 2.32 | 1.46–3.68 | <0.001 |
| Mechanical Ventilation | | | <0.001 (LR test) |
| Phenotype I | 25.59 | 7.69–85.17 | <0.001 |
| Phenotype II | 7.45 | 2.27–24.43 | <0.001 |
| **Count Outcome** *(Binomial Regression)* | IRR | 95% CI | P value |
| Hospital LOS* | | | <0.001 (LR test) |
| Phenotype I | 1.74 | 1.45–2.10 | <0.001 |
| Phenotype II | 1.22 | 1.05–1.43 | 0.01 |

Abbreviations: PH, proportional hazards; HR, hazard ratio; CI, confidence interval; OR, odds ratio; ICU, intensive care unit; IRR, incidence rate ratio; LOS, length of stay; LR, likelihood ratio.

Reference group for all models is Phenotype III. All models adjusted for sex, race/ethnicity, and Elixhauser Comorbidity Index.

* LOS model only included patients that survived.

socioeconomic status was similar across all phenotypes, which has been proposed to be a driver of disparate outcomes in healthcare. These findings are consistent with a recent study conducted across this populations of patients which found COVID-19 severity to be associated with minority populations and non-English speaking patients, independent of socioeconomic status. Given race/ethnicity and primary language spoken are social constructs and traits, respectively, which are not biologically grounded; these results require further investigation as to why these populations are at higher risk of developing phenotype I through mediation analysis of external factors (as opposed to these populations being an isolated cause of developing an unfavorable phenotype).

Phenotype III was associated with the best clinical outcomes and can be termed the "Favorable Phenotype". Surprisingly, patients with phenotype III had a very high rate of respiratory comorbidities and the best clinical outcomes. What is most surprising is despite the lowest complication rate and mortality, this phenotype was associated with a greater than 10% rate of hospital readmission. Long-term sequelae from the critically ill remains an important target for patient centered improvements in care given the increasing loss of functional status among ICU patients predating the pandemic. It is possible that patients pre-existing respiratory comorbidities predisposed them to longer term sequelae which may have resulted in this readmission rate, although additional studies are needed to better elucidate these findings, specifically controlling for differences in survival. Patients with respiratory comorbidities such as asthma and COPD routinely use medications which may be protective in SARS-CoV-2 pathogenesis which may explain this protective effect. For example, our group has previously identified reduced mortality in COVID-19 for patients with asthma treated with beta2-agonists [16]. Patients with phenotype III were more likely to use inhaled steroids, nasal fluticasone, albuterol, and antihistamines.

Clinical phenotypes are critical during a pandemic when time and resources are scarce. Phenotypes not only enable the identification of risk factors; they also provide essential insight

towards the high yield follow up investigations. For example, while noting the respiratory associations with phenotype III (favorable phenotype) is interesting, the more beneficial take away includes further investigations towards how these underlying conditions and/or their medications may mitigate illness severity. Lastly, by phenotyping patients affected by COVID-19; we set the foundation to begin comparing if these phenotypes are unique to SARS-CoV-2 or if similarities exist elsewhere.

As the attention paid to personalized medicine accelerates; these studies are just the beginning. Future work will expand upon these phenotypes with the hope that they can assist in 1) identifying those at risk of poor outcomes, 2) precisely treating each phenotype (which may not be uniform across all phenotypes), and 3) preventing further complications in those phenotypes at higher risk. In addition, a deeper investigation into clinical phenotypes and associated genomic, transcriptomic, and proteomic is needed. The ability to classify patients into clinical phenotypes can facilitate the linkage of—omics data to better understand SARS-CoV-2 pathogenesis and natural history. Work is already being done to identify genetic host factors that may play a role in determining not only susceptibility to the virus, but also the clinical trajectory when infection does occur. Understanding COVID-19 severity, its biomarkers, and risk factors is paramount during the COVID-19 pandemic.

Our study has several limitations, including that this is a retrospective study and therefore results may be biased or subject to residual confounding. Second, patients were followed for variable lengths of time. Patients that were admitted in March 2020 thus had approximately 5 months of follow-up whereas patients admitted in late August had limited time. We accounted for this by conducting a Cox proportional hazard analysis when analyzing in- and out- of hospital mortality. Additionally, when the data were pulled, only 54 patients (5%) remained hospitalized. While most patients developed complications within their first 2 weeks of hospital admission, it is possible that they may still develop clinical complications which is not reflected in this analysis. Furthermore, our analysis was completed on hospitalized patients. It is important to recognize that our results are restricted to those who required hospitalization. Our data cannot be extrapolated to those with mild COVID-19 (i.e. not requiring hospitalization).

## Conclusion

In this retrospective analysis of patients with COVID-19, three clinical phenotypes were identified reflecting adverse, moderate, and favorable outcomes. Patients from each phenotype presented with different comorbidities and developed different complications. Our results suggest that phenotype-specific medical care of COVID-19 may improve outcomes. Future research is urgently needed to determine the utility of these phenotypes in clinical practice and trial design.

## Supporting information

**S1 Fig. Consensus cumulative distribution functions.** Cumulative distribution functions (CDF) for a randomly selected imputed dataset are shown. A range of phenotypes (2–7) were considered, and the optimal choice of phenotypes is 3.
(TIF)

**S2 Fig. Delta area.** The relative change in delta area under the cumulative distribution function is shown for the range of phenotypes (k = 2–7) for a randomly selected imputed dataset. The optimal choice of phenotypes is 3. Abbreviations: CDF (cumulative distribution function).
(TIF)

**S3 Fig. Consensus matrix with 3 clusters.** A consensus matrix heatmap is shown for a randomly selected imputed dataset clustered into 3 phenotypes. The heatmap allows visualization of consensus cluster assignments to evaluate cluster stability. Darker shades of green indicate higher stability.
(TIF)

**S4 Fig. Consensus matrix with 4 clusters.** A consensus matrix heatmap is shown for a randomly selected imputed dataset clustered into 4 phenotypes. The heatmap allows visualization of consensus cluster assignments to evaluate cluster stability. Darker shades of green indicate higher stability. The choice of 4 clusters shows less stability than 3 clusters (see S3 Fig).
(TIF)

**S5 Fig. Cumulative proportion of variance explained.** The proportion of variance explained by each principal component is summed over all principal components. For example, PC1 and PC2 cumulatively explain 20% of the variation in the dataset. Abbreviations: PC1 (principal component 1); PC2 (principal component 2).
(TIF)

**S6 Fig. Contribution of variables to PC1.** The contributions of each of the 33 variables used in the clustering to principal component 1 are shown. The red line marks the expected average contribution of each variable if the contributions of the variables were uniform across the dataset. Variables contributing most to the observed pattern in PC1 are D-dimer and albumin. Abbreviations: PC1 (principal component 1); Abs_Nphil_Ct (absolute neutrophil count); LDH (lactate dehydrogenase); CRP (C-reactive protein); WBC (white blood cell count); HCT (hematocrit); HGB (hemoglobin); Tbili (total bilirubin); RDW (red cell distribution width); AST (aspartate aminotransferase); Alk_phos (alkaline phosphatase); RR (respiratory rate); CA (calcium); TP (total protein); INR (internal normalized ratio of prothrombin time); CO2 (carbon dioxide); K (potassium); O2SAT (oxygen saturation); BMI (body mass index); PLT (platelet); PP (pulse pressure); Na (sodium); SBP (systolic blood pressure); Abs_mono_ct (absolute monocyte count); MCV (mean corpuscular volume).
(TIF)

**S7 Fig. Comorbidities by phenotype.** Chord diagram illustrates the prevalence of comorbidities (% observed) for the three clinical phenotypes.
(TIF)

**S8 Fig. Complications by phenotype.** Chord diagram illustrates the prevalence of complications (% observed) for the three clinical phenotypes.
(TIF)

**S9 Fig. Clinical outcomes by phenotype.** Chord diagram illustrates the prevalence of clinical outcomes (% observed) for the three clinical phenotypes. Abbreviations: ICU (intensive care unit); Vent (mechanical ventilation); Readmit (readmission to hospital or ICU); ECMO (extracorporeal membrane oxygenation).
(TIF)

**S1 Table. Categories of comorbidities and ICD 10 codes used.**
(PDF)

**S2 Table. List of complications contributing to each complication category.**
(PDF)

**S3 Table. Association of clinical phenotype with in-hospital complications.**
(PDF)

**S4 Table.**
(PDF)

**S5 Table. Home medications and hospital day 5 laboratory values of hospitalized COVID-19 patients with clinical phenotypes I, II, and III.**
(PDF)

**S1 File.**
(XLSX)

## Author Contributions

**Conceptualization:** Elizabeth R. Lusczek, Nicholas E. Ingraham, Basil S. Karam, Jennifer Proper, Lianne Siegel, Erika S. Helgeson, Sahar Lotfi-Emran, Emily J. Zolfaghari, Emma Jones, Michael G. Usher, Jeffrey G. Chipman, R. Adams Dudley, Bradley Benson, Genevieve B. Melton, Anthony Charles, Monica I. Lupei, Christopher J. Tignanelli.

**Data curation:** Elizabeth R. Lusczek, Nicholas E. Ingraham, Christopher J. Tignanelli.

**Formal analysis:** Elizabeth R. Lusczek, Nicholas E. Ingraham, Jennifer Proper, Lianne Siegel, Erika S. Helgeson, Christopher J. Tignanelli.

**Writing – original draft:** Elizabeth R. Lusczek, Nicholas E. Ingraham, Basil S. Karam, Jennifer Proper, Lianne Siegel, Erika S. Helgeson, Sahar Lotfi-Emran, Emily J. Zolfaghari, Emma Jones, Michael G. Usher, Jeffrey G. Chipman, R. Adams Dudley, Bradley Benson, Genevieve B. Melton, Anthony Charles, Monica I. Lupei, Christopher J. Tignanelli.

**Writing – review & editing:** Elizabeth R. Lusczek, Nicholas E. Ingraham, Basil S. Karam, Jennifer Proper, Lianne Siegel, Erika S. Helgeson, Sahar Lotfi-Emran, Emily J. Zolfaghari, Emma Jones, Michael G. Usher, Jeffrey G. Chipman, R. Adams Dudley, Bradley Benson, Genevieve B. Melton, Anthony Charles, Monica I. Lupei, Christopher J. Tignanelli.

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
