## [Decision Letter · Decision Letter 0]

31 Dec 2020

PONE-D-20-32255

Characterizing COVID-19 Clinical Phenotypes and Associated Comorbidities and Complication Profiles

PLOS ONE

Dear Dr. Ingraham,

Thank you for submitting your manuscript to PLOS ONE. After careful consideration, we feel that it has merit but does not fully meet PLOS ONE’s publication criteria as it currently stands. Therefore, we invite you to submit a revised version of the manuscript that addresses the points raised during the review process.

We look forward to receiving your revised manuscript.

Kind regards,

Chiara Lazzeri

Academic Editor

PLOS ONE

Journal Requirements:

2. In the ethics statement, please provide further clarification whether the IRB of all participating hospitals provided ethical approval.

3. Please note that all PLOS journals ask authors to adhere to our policies for sharing of data and materials: https://journals.plos.org/plosone/s/data-availability. According to PLOS ONE’s Data Availability policy, we require that the minimal dataset underlying results reported in the submission must be made immediately and freely available at the time of publication. As such, please remove any instances of 'unpublished data' or 'data not shown' in your manuscript and replace these with either the relevant data (in the form of additional figures, tables or descriptive text, as appropriate), a citation to where the data can be found, or remove altogether any statements supported by data not presented in the manuscript.

4. We noted in your submission details that a portion of your manuscript may have been presented or published elsewhere.

"It has been published as a preprint: " ext-link-type="uri" xlink:type="simple">https://www.medrxiv.org/content/10.1101/2020.09.12.20193391v1"

Please clarify whether this publication was peer-reviewed and formally published. If this work was previously peer-reviewed and published, in the cover letter please provide the reason that this work does not constitute dual publication and should be included in the current manuscript.

7. Your ethics statement should only appear in the Methods section of your manuscript. If your ethics statement is written in any section besides the Methods, please delete it from any other section.

Reviewers' comments:

Reviewer's Responses to Questions

**Comments to the Author**

1. Is the manuscript technically sound, and do the data support the conclusions?

Reviewer #1: Yes

Reviewer #2: Yes

2. Has the statistical analysis been performed appropriately and rigorously? 

Reviewer #1: I Don't Know

Reviewer #2: Yes

3. Have the authors made all data underlying the findings in their manuscript fully available?

Reviewer #1: Yes

Reviewer #2: Yes

4. Is the manuscript presented in an intelligible fashion and written in standard English?

Reviewer #1: Yes

Reviewer #2: Yes

5. Review Comments to the Author

Reviewer #1: Follow the tips in the attachment.

Complete the description of the tables.

Follow the journal format.

Mention the study interval.

You need to explain the practical purpose further.

Identify the importance of the studied phenotypes and intermediate phenotypes.

Reviewer #2: The Authors report on a retrospective study aimed at assessing whether distinct phenotypes can be identified within the COVID-19 spectrum of clinical presentation. A large series of patients admitted to 14 hospital was ascertained.

As the Authors sensibly state, a multidimensional approach is needed to better understand COVID-19 and interpret the variation in clinical outcomes. The article may provide novel pieces of evidence in order to establish a reliable stratification of patients.

The Authors reported that the data set includes all consecutive patients – this strategy limits the ascertainment bias and is an element of strength of the study.

A few issues could be addressed to improve the overall quality of the manuscript.

The study cohort comprised patients admitted to inpatient clinics (n=1022, out of 7538 patients). Therefore, the wide spectrum of phenotypes caused by SARS-CoV-2 infection which was exhibited by the majority of individuals could not be accounted for. I understand that the recruitment setting is constrained by the study design. However, the generalisability of results should be discussed accounting for this limitation.

To put data into the health care context, a brief description of the hospital setting could be helpful – e.g. geographical distribution, dimension of the hospital, type of unit [if not ICU], etc., including the population served.

It is noteworthy that phenotype I was found associated with being non-white and non-English speaking. Though the socioeconomic status was not differently distributed, this finding should be discussed.

To this regard, it should be reported how race/ethnicity was ascertained.

Dissecting the role for constitutional risk factors, and particularly genetic risk factors, is of paramount importance to design effective health care strategies for COVID-19. To this purpose, a clear-cut, evidence-based characterisation of phenotypes is a fundamental step. With this perspective, the implications of the present study deserve to be properly addressed.

Conversely, the Authors outlined the impact of the study in a very simplistic way.

As far as concerns genetic predisposition, accelerated aging is far from being a pivotal reference [page 10 first paragraph, ref. 30]; the term ‘exome data’ [page 11, last paragraph of Discussion] is inappropriate. The Authors should be aware that there is a line of research focussing on the role for host genetic factors in determining variable susceptibility to develop the phenotypes associated to SARS-CoV-2 infection. A large body of literature has been published – see for instance ‘Genetic variants of the human host influencing the coronavirus-associated phenotypes (SARS, MERS and COVID-19): rapid systematic review and field synopsis’, Human Genomics 2020, which also addresses the quality of methodological approaches; Beck and Aksentijevich, Science 2020, and citations therein; and the recently published genome-wide association studies.

Minor issues:

- Some references are incomplete.

- A few acronyms should be defined [e.g. SOFA, RDW].

6. PLOS authors have the option to publish the peer review history of their article (what does this mean?). If published, this will include your full peer review and any attached files.

Reviewer #1: No

Reviewer #2: No

---

## [Author Response · Author response to Decision Letter 0]

19 Feb 2021

The response to reviewers has also been uploaded with in a word format with out manuscript. Below includes the responses as well.

01.Response: The manuscript has been appropriately formatted to match PLOS ONE’s style requirements.

2. In the ethics statement, please provide further clarification whether the IRB of all participating hospitals provided ethical approval.

02.Response: The IRB was approved by all hospitals within the MHealth Fairview system which includes ethical approval. This has been updated in the manuscript.

3. Please note that all PLOS journals ask authors to adhere to our policies for sharing of data and materials: https://journals.plos.org/plosone/s/data-availability. According to PLOS ONE’s Data Availability policy, we require that the minimal dataset underlying results reported in the submission must be made immediately and freely available at the time of publication. As such, please remove any instances of 'unpublished data' or 'data not shown' in your manuscript and replace these with either the relevant data (in the form of additional figures, tables or descriptive text, as appropriate), a citation to where the data can be found, or remove altogether any statements supported by data not presented in the manuscript.

03.Response: Thank you for pointing this out, we have added a table which includes the data from the sensitivity analysis. We have also updated the numbering of our supplemental figures to the appropriate order. 

4. We noted in your submission details that a portion of your manuscript may have been presented or published elsewhere.

"It has been published as a preprint: https://www.medrxiv.org/content/10.1101/2020.09.12.20193391v1"

Please clarify whether this publication was peer-reviewed and formally published. If this work was previously peer-reviewed and published, in the cover letter please provide the reason that this work does not constitute dual publication and should be included in the current manuscript.

04.Response: This publication was not peer-reviewed, and the cover letter has been updated to convey this as well. 

05.Response: We have put together a minimal database to be included with our manuscript to support the public data sharing. 

06.Response: This has been addressed in Response #3. In brief, we have added a figure to ensure all of our data is included in the manuscript.

7. Your ethics statement should only appear in the Methods section of your manuscript. If your ethics statement is written in any section besides the Methods, please delete it from any other section.

07.Response: This has been corrected.

08.Response: This has been corrected.

Reviewers' comments:

Reviewer's Responses to Questions

5. Review Comments to the Author

Reviewer #1: 

Follow the tips in the attachment.

1) Complete the description of the tables.

09.Response: Thank you for this feedback. This has been corrected in the manuscript

2) Follow the journal format.

10.Response: The manuscript has been updated to the appropriate format for PLOS ONE.

3) Mention the study interval.

11.Response: Thank you for this comment. The study interval is listed in the methods section of the manuscript; however, we have also added it to the abstract March 7, 2020 to August 25, 2020

4) You need to explain the practical purpose further. Identify the importance of the studied phenotypes and intermediate phenotypes.

12.Response: This is a great point and we thank you for the comment. We have expanded upon our discussion regarding the utility of these phenotypes, clinically and academically. (Last paragraph of page 14 through top of page 15)

Reviewer #2: 

The Authors report on a retrospective study aimed at assessing whether distinct phenotypes can be identified within the COVID-19 spectrum of clinical presentation. A large series of patients admitted to 14 hospitals was ascertained.

As the Authors sensibly state, a multidimensional approach is needed to better understand COVID-19 and interpret the variation in clinical outcomes. The article may provide novel pieces of evidence in order to establish a reliable stratification of patients.

The Authors reported that the data set includes all consecutive patients – this strategy limits the ascertainment bias and is an element of strength of the study.

A few issues could be addressed to improve the overall quality of the manuscript.

1) The study cohort comprised patients admitted to inpatient clinics (n=1022, out of 7538 patients). Therefore, the wide spectrum of phenotypes caused by SARS-CoV-2 infection which was exhibited by the majority of individuals could not be accounted for. I understand that the recruitment setting is constrained by the study design. However, the generalisability of results should be discussed accounting for this limitation.

13.Response: We appreciate your feedback. This is an important point and we have updated our limitations section to include this point. 

To put data into the health care context, a brief description of the hospital setting could be helpful – e.g. geographical distribution, dimension of the hospital, type of unit [if not ICU], etc., including the population served.

It is noteworthy that phenotype I was found associated with being non-white and non-English speaking. Though the socioeconomic status was not differently distributed, this finding should be discussed.

To this regard, it should be reported how race/ethnicity was ascertained.

14.Response: This is a great point. We have expanded upon the hospital system characteristics. Regarding the racial/ethnic and primary language association, we very much appreciate this point. We are currently looking into this and have a manuscript under review which further supports these findings (Race/Ethnicity + Non-English-speaking populations are associated with a higher rate of severe COVID-19 disease which is all independent of socioeconomic status). We have added more to the paragraph regarding phenotype I and made a point to distinguish that this association should be viewed appropriately given the social construct in which race and ethnicity are derived. 

Dissecting the role for constitutional risk factors, and particularly genetic risk factors, is of paramount importance to design effective health care strategies for COVID-19. To this purpose, a clear-cut, evidence-based characterisation of phenotypes is a fundamental step. With this perspective, the implications of the present study deserve to be properly addressed.

Conversely, the Authors outlined the impact of the study in a very simplistic way.

As far as concerns genetic predisposition, accelerated aging is far from being a pivotal reference [page 10 first paragraph, ref. 30]; the term ‘exome data’ [page 11, last paragraph of Discussion] is inappropriate. The Authors should be aware that there is a line of research focussing on the role for host genetic factors in determining variable susceptibility to develop the phenotypes associated to SARS-CoV-2 infection. A large body of literature has been published – see for instance ‘Genetic variants of the human host influencing the coronavirus-associated phenotypes (SARS, MERS and COVID-19): rapid systematic review and field synopsis’, Human Genomics 2020, which also addresses the quality of methodological approaches; Beck and Aksentijevich, Science 2020, and citations therein; and the recently published genome-wide association studies.

15.Response: Thank you for this great insight. We have strengthened our section which describes the implications and importance of our work. We have also included the suggested citations (much appreciated) which further drives home our point. 

Minor issues:

- Some references are incomplete.

16.Response: This has been corrected

- A few acronyms should be defined [e.g. SOFA, RDW].

17.Response: This has been corrected

---

## [Editor Report · Decision Letter 1]

9 Mar 2021

Characterizing COVID-19 Clinical Phenotypes and Associated Comorbidities and Complication Profiles

PONE-D-20-32255R1

Dear Dr. Ingraham,

We’re pleased to inform you that your manuscript has been judged scientifically suitable for publication and will be formally accepted for publication once it meets all outstanding technical requirements.

Kind regards,

Chiara Lazzeri

Academic Editor

PLOS ONE
---

## [Editor Report · Acceptance letter]

18 Mar 2021

PONE-D-20-32255R1 

Characterizing COVID-19 clinical phenotypes and associated comorbidities and complication profiles 

Dear Dr. Ingraham:

I'm pleased to inform you that your manuscript has been deemed suitable for publication in PLOS ONE. Congratulations! Your manuscript is now with our production department. 

Kind regards, 

on behalf of

Dr. Chiara Lazzeri 

Academic Editor

PLOS ONE